# Antibacterial and Osteogenic Properties of Ag Nanoparticles and Ag/TiO_2_ Nanostructures Prepared by Atomic Layer Deposition

**DOI:** 10.3390/jfb13020062

**Published:** 2022-05-18

**Authors:** Denis Nazarov, Ilya Ezhov, Natalia Yudintceva, Maxim Shevtsov, Aida Rudakova, Vladimir Kalganov, Vladimir Tolmachev, Yuliya Zharova, Oleksiy Lutakov, Ludmila Kraeva, Elizaveta Rogacheva, Maxim Maximov

**Affiliations:** 1Saint Petersburg State University, Universitetskaya nab, 7/9, 199034 Saint Petersburg, Russia; aida.rudakova@spbu.ru (A.R.); vdkalganov@yandex.ru (V.K.); 2Peter the Great Saint Petersburg Polytechnic University, Polytechnicheskaya, 29, 195221 Saint Petersburg, Russia; iezhov1994@gmail.com (I.E.); maximspbstu@mail.ru (M.M.); 3Institute of Cytology of the Russian Academy of Sciences (RAS), Tikhoretsky Ave., 4, 194064 Saint Petersburg, Russia; yudintceva@mail.ru (N.Y.); shevtsov-max@mail.ru (M.S.); 4Center of Translational Cancer Research (TranslaTUM), Klinikum Rechts der Isar, Technical University Munich, Einstein Str. 25, 81675 Munich, Germany; 5Ioffe Institute, Polytechnicheskaya, 26, 194021 Saint Petersburg, Russia; tva@mail.ioffe.ru (V.T.); piliouguina@mail.ioffe.ru (Y.Z.); 6Department of Solid State Engineering, Institute of Chemical Technology, 16628 Prague, Czech Republic; oleksiy.lyutakov@vscht.cz; 7Saint-Petersburg Pasteur Institute of Epidemiology and Microbiology, 14 Mira Street, 197101 Saint Petersburg, Russia; lykraeva@yandex.ru (L.K.); elizvla@yandex.ru (E.R.)

**Keywords:** silver, titania, nanoparticles, thin films, atomic layer deposition, cell viability, cell differentiation, mesenchymal stromal cells, antibacterial properties

## Abstract

The combination of titania nanofilms and silver nanoparticles (NPs) is a very promising material, with antibacterial and osseointegration-induced properties for titanium implant coatings. In this work, we successfully prepared TiO_2_ nanolayer/Ag NP structures on titanium disks using atomic layer deposition (ALD). The samples were studied by scanning electron microscopy (SEM), X-ray diffraction, X-ray photoelectron spectroscopy (XPS), contact angle measurements, and SEM-EDS. Antibacterial activity was tested against *Staphylococcus aureus*. The in vitro cytological response of MG-63 osteosarcoma and human fetal mesenchymal stem cells (FetMSCs) was examined using SEM study of their morphology, MTT test of viability and differentiation using alkaline phosphatase and osteopontin with and without medium-induced differentiation in the osteogenic direction. The samples with TiO_2_ nanolayers, Ag NPs, and a TiO_2_/Ag combination showed high antibacterial activity, differentiation in the osteogenic direction, and non-cytotoxicity. The medium for differentiation significantly improved osteogenic differentiation, but the ALD coatings also stimulated differentiation in the absence of the medium. The TiO_2_/Ag samples showed the best antibacterial ability and differentiation in the osteogenic direction, indicating the success of the combining of TiO_2_ and Ag to produce a multifunctional biocompatible and bactericidal material.

## 1. Introduction

To date, significant progress has been made in improving the mechanical properties of titanium implants. The development of new titanium alloys [1,2] and the nanostructuring of pure titanium [3] have significantly increased the strength, hardness, and wear resistance of the material, which brings the elasticity modulus closer to the properties of bone tissue, and thereby improves its biocompatibility. However, the current development of new titanium-based implants has shifted the emphasis from favorable mechanical properties to improved biocompatibility, antibacterial properties, absence of cytotoxicity and, finally, fast and successful osseointegration. The issue of biocompatibility is solved by modifying the surface of the material, such as coating, biofunctionalization, creating a certain surface topography by physical and chemical methods, etc. [4]. However, achieving all of the above characteristics of the material is an extremely difficult task. One of the most challenging issues is the combination of the antibacterial effect of the implant surface with a positive cellular response and accelerated osseointegration.

The biocompatibility of titanium implants is due to the presence of a stable and non-toxic oxide on their surfaces. There are many works that show that crystalline titanium oxide stimulates cells adhesion, proliferation, and crystallization of the resulting bone tissue and promotes rapid osseointegration [5,6,7,8]. Thus, the application of crystalline titanium oxide coatings is one of the simplest and, at the same time, very effective ways to improve the osseointegration of the material. Despite the great progress in the development of new bioactive materials with accelerated osseointegration, the problem of a large number of implant rejections is still very relevant. A 5-year retrospective study (2015–2019) of 6113 dental implants indicated that the rate of failure is about 1.2% [9]. Most of these cases were caused by bacterial infection. It has also been reported that deep infections typically occur in 1–2% of patients with hip arthroplasties [10], causing an acute and chronic inflammatory reaction, osteolysis, loosening, and implant failure. These cases often require implant removal and reimplantation, which causes additional inconvenience to the patients and financial expenses. The main approach to solving this problem focuses on the simultaneous action of antibacterial and bioactive coatings.

Advances in the development of new antibiotics have solved many medical complications. However, the problem of multiple antibiotic resistances of new strains does not allow us to abandon the use of many classic antibacterial materials, such as silver. Silver nanoparticles (NPs) have proved to be effective in medical implants because of their high surface area and antibacterial properties against Gram-positive and Gram-negative bacteria [11]. The antimicrobial effect of silver NPs is achieved via both the contact-killing mechanism (direct contact bacteria with Ag) or release-killing mechanism due to the release of oxidative Ag ions from the NPs, formation of reactive oxygen species (ROS) or inactivation of respiratory enzymes by reacting with sulfhydryl groups on the surface [12].

Despite the above, the practical use of silver as an antibacterial material is associated with a number of problems. The poor reproducibility of antibacterial activity of Ag-containing surfaces has been revealed in many studies [13]. In addition, for thick silver coatings and particles (>1 μm), a strong long-term cytotoxic effect is observed. For NPs, despite the high specific surface area and, therefore, high ion release rate, the long-term cytotoxic effect should be lower. In this regard, the question of the balance of antibacterial properties, biocompatibility and osteoinductivity when using silver NPs remains quite relevant.

In this study, we obtained a continuous titanium oxide layer decorated with silver nanoparticles and evaluated their combined effect on antibacterial properties and cytological response. Atomic layer deposition (ALD) technology was used to synthesize a composite of titanium oxide nanocoatings and silver NPs.

ALD is a chemical technology for producing thin films and coatings based on cyclic and self-limiting reactions at the gas–solid interface. During the ALD, pairs of reagents are sequentially chemisorbed on the substrate surface [14]. The cyclical nature of the ALD processes provides precise thickness control down to the sub-nanometer. In addition, self-limiting surface reactions at the substrate–gas interface provide layer-by-layer film growth and allow for conformal deposition of thin films on complex three-dimensional and porous substrates. These features and good scalability have led to an increase in industry interest for ALD and a dramatic increase in publications over the last decade [15]. The possibility of using one technology for the synthesis of both continuous coatings of titanium oxide and silver NPs is also an important factor when choosing ALD as a method for the preparation of biomedical coatings.

In the ALD processes, the use of suitable precursors is critical. For the deposition of titanium oxide, titanium chloride and water were chosen as a titanium-containing precursor and a co-reagent, respectively. This reagent pair is standard for the ALD of titanium oxide and allows the deposition of pure polycrystalline films with the anatase structure at temperatures below 200 °C. To date, many precursors have been used for Ag deposition by ALD, but the optimal one has not yet been found. Most silver-containing ALD precursors are non-volatile substances that are unstable at standard ALD temperatures (150–200 °C) [16]. Among them, the most stable and affordable precursor (2,2-dimethyl-6,6,7,7,8,8,8-heptafluorooctane-3,5-dionato) silver(I) triethyl-phosphine (Ag(fod)(PEt_3_)–C_16_H_25_AgF_7_O_2_P) was chosen. Hydrogen plasma was used as the reducing agent. It should be noted that the ALD synthesis of silver with a relatively small number of cycles produces individual nanoparticles rather than a conformal layer [17]. As the number of cycles increases, the particles increase in their size and merge with each other to form a continuous layer.

## 2. Materials and Methods

### 2.1. Samples Preparation

The ultrafine-grained (UFG) titanium discs (diameter 6 mm) and monocrystalline silicon (diameter 100 mm) were used as supports. UFG Ti raw material was prepared in Nanomet LLC (Ufa, Russia) from the grade 4 titanium. Titanium rods of 1 m length were subjected to Equal-Channel Angular Pressing by ECAP-Conform processing at 400 °C, as described elsewhere [18]. After processing, the billets drawing at 200 °C resulted in the production of UFG rods with a grain size of about 200–300 nm.

The obtained UFG rods were treated by machining as described elsewhere [19,20]. First, the rods were cut into discs (thickness of 2–3 mm) with the Buehler IsoMet 1000 machine (Buehler, Lake Bluff, IL, USA). Then, the discs were ground and polished by a semiautomatic Buehler MiniMet 1000 machine (Buehler, Lake Bluff, IL, USA) to a mirror-like surface using 200, 400, 800, and 1200 grit sandpapers and silicon dioxide nanoparticles suspension (50 nm). Prior to etching, the samples were cleaned repeatedly with acetone and deionized water in an ultrasonic bath for 15 min and dried in an argon flow.

### 2.2. Atomic Layer Deposition

Titanium oxide coatings were prepared by ALD on the surface of the polished UFG-Ti and monocrystalline silicon (100) plates (2 × 2 cm). Silicon wafers acted as a witness (control sample) for ellipsometry and X-ray reflectometry (XRR). The deposition was performed in the hot-wall, flow-type reactor (Nanoserf) with slot-type geometry (Nanoengineering Ltd., Saint Petersburg, Russia). The temperature of the reactor was maintained at 200 °C. Nitrogen (99.9999%) was used as a carrier and purging gas. The reactor gas flow rate at purge pulse was 300 sccm (standard cubic centimeters per minute). Titanium chloride (TiCl_4_, 99%, Merck, NJ, USA) and high-purity water were used as precursors. The pulse duration was 100 ms for both reagents. The total number of TiO_2_ ALD cycles was 400.

The Ag nanoparticles were deposited by ALD with a Picosun R-150 setup using (2,2-dimethyl-6,6,7,7,8,8,8-heptafluorooctane-3,5-dionato) silver(I) triethyl-phosphine (Ag(fod)(PEt_3_), C_16_H_25_AgF_7_O_2_P) (98%, Strem Chemicals, Newburyport, MA, USA) as the Ag-containing precursor and hydrogen plasma as a reducing agent. To determine the optimal growth conditions, the reactor temperatures (142–184 °C), (Ag(fod)(PEt_3_)) evaporator temperatures (115–150 °C), reagents pulse times (2–4 s) and number of pulses (1–11) in one ALD cycle were varied. The total number of Ag ALD cycles was varied from 350 to 2300.

### 2.3. Samples Characterization

The thickness of Ag and TiO_2_ coatings deposited on the silicon witnesses was estimated by spectroscopic ellipsometry (350–1000 nm) using an Ellips-1891 SAG instrument (CNT, Novosibirsk, Russia). The accuracy of the thickness measurement was estimated as 0.5–1 nm. The samples were imaged with the scanning electron microscope Zeiss Merlin operated at 20 kV. Microscope spatial resolution was around 2 nm and magnification up to 300,000. SE (secondary electrons), In-lens and EDS regimes were used.

X-ray reflectometry (XRR) and X-ray diffraction (XRD) studies in surface sensitive grazing incidence XRD (GIXRD) modes were performed using a Bruker D8 DISCOVER (Cu-Ka) high-resolution diffractometer. XRD studies were conducted in the 20–85° range with a 0.05° step. XRR measurements were made in the range of angles from 0.31° to 5° with the increment 0.01 using symmetric scattering geometry. The obtained results were processed by the Rietveld method using TOPAS 5 software (XRD) and by the simplex method using LEPTOS 7.7 (XRR).

The chemical composition of the sample surface was studied by X-ray photoelectron spectroscopy (XPS). XPS spectra were measured with a Thermo Fisher Scientific Escalab 250Xi photoelectron spectrometer. The samples were excited by Al Kα (1486.7 eV) X-rays. An ion gun was used for surface sputtering. Survey spectra and high resolution C1s, O1s, Ti2p, Ag3d, F1s, P2p spectra were registered.

The contact angle (CA) measurements were carried out by the sessile drop method using a Theta Lite optical tensiometer (Biolin Scientific, Gothenburg, Sweden) as described elsewhere [21]. One Attension Software was used for processing the results. The mean CA values (Θ) were calculated using measurements from 5 identical samples at 2–3 surface points. The volume of liquid droplets was no more 2.2 µL. The surface free energy (SFE) values were calculated by the Owens–Wendt–Rabel–Kaelble (OWRK)/Fowkes approach using the two-liquid method (water versus diiodomethane-DIM) [22]. Ultrapure water served as a liquid with a dominant polar component of SFE (SFE_p_ = 51.0 mN/m, SFE_d_ = 21.8 mN/m), and diiodomethane (99%, stabilized by copper, Sigma-Aldrich) was used as a liquid with a dominant dispersive component of SFE (SFE_d_ = 48.5 mN/m, SFE_p_ = 2.3 mN/m).

### 2.4. In Vitro Assessment of the Cellular Response

#### 2.4.1. Cell Culture

Human osteosarcoma cell line (MG-63) (ATCC^®^ CRL-1427TM) and human fetal mesenchymal stem cells derived from bone marrow (FetMSCs) were obtained from the shared research facility “Vertebrate cell culture collection”, supported by the Ministry of Science and Higher Education of the Russian Federation (Agreement № 075-15-2021-683). MG-63 and FetMSCs were maintained in a CO_2_-incubator (37 °C, 6% CO_2_) in DMEM medium (Sigma-Aldrich, Burlington, MA, USA) supplemented with 2 mM of L-glutamine, 10% fetal bovine serum (FBS) and gentamicine (Gibco, Thermo Fisher Scientific, Waltham, MA, USA).

#### 2.4.2. Cell Morphology

To sterilize the titanium samples, they were autoclaved in water steams at 121 °C and a pressure of 1.5 atmospheres. Then, the samples were placed into the wells of 4-well plates (Nunc, USA). FetMSCs and MG-63 cells were seeded (1 × 10^5^ in 20 µL of DMEM/F12 nutrient medium) on the surface of the samples and maintained for 24 h in CO_2_. After incubation, the cells were washed with Dulbecco’s phosphate buffer saline (PBS) (Sigma-Aldrich, Burlington, MA, USA) and fixed in 2.5% glutaraldehyde in phosphate buffer (pH = 7.2, Sigma-Aldrich, Burlington, MA, USA). After 3 days, the samples were washed in a phosphate buffer and successively dehydrated in 30, 50, 70, 90, 96% and absolute ethanol for 30 min each. The final drying was carried out 3 times for 15 min using Leica EM CPD300 at the CO_2_ critical point. Finally, the Au coatings with a thickness of about 10 nm were deposited with Leica EM SCD500. The evaluation of the cells’ morphology was performed using a Tescan MIRA3 LMU scanning electron microscope (TESCAN, Brno–Kohoutovice, Czech Republic).

#### 2.4.3. Cell Viability

The suspensions of MG-63 and FetMSCs were seeded into wells of a 96-well plate (Nunc, USA) at the concentration of 5000 cells/well. Three titanium samples with an area of 0.8 cm^2^ were incubated in 1 mL of nutrient medium for 24 h at 37 °C. The conditioned medium was used to assess the cytotoxicity of the material on cells. Standard cell culture condition was used as a control. The cytotoxicity was assessed after 24 h using the Vibrant MTT Cell Proliferation Assay Kit (Life Technologies, Carlsbad, CA, USA) according to the manufacturer’s protocol.

#### 2.4.4. Cells Osteogenic Differentiation Analysis

For the evaluation of the cells’ osteogenic differentiation, an early marker alkaline phosphatase (ALP) and a late marker osteopontin (OPN) were analyzed [23]. The cell seedings were carried out in the same way as for the morphology study. After seeding, 300 µL of nutrient medium was added to each well to completely cover the sample surface. The assessment was performed after 7, 14 and 21 days of cells co-incubation on samples in a CO_2_-incubator with and without the cell differentiation-induced medium (StemPro Osteogenesis Differentiation Kit, Thermofisher Scientific). The medium was changed once a week. As a control, we used standard conditions for culturing cells on the surface of the culture plastic. The medium was sampled at each medium change and placed at −80 °C for subsequent assessment of the presence of ALP and OPN using the Human Alkaline Phosphatase Assay ELISA Kit (Thermo Fisher Scientific, Waltham, MA, USA) and the Osteopontin/SSP1 Human ELISA Kit (Thermo Fisher Scientific, USA), according to the manufacturer’s protocol. Optical density measurements were performed on a Thermo Scientific Varioskan LUX multimodal reader at a wavelength of 450 nm.

#### 2.4.5. Statistical Analysis

Three and five samples of each type were used for MTT test and differentiation analysis, respectively. The error bars in the figures represent the confidence interval (CI). Student’s *t* tests were used to evaluate the differences between the experimental and control groups. A credible interval of *p* < 0.05 was considered statistically significant for all the tests.

### 2.5. Antibacterial Activity

The assessment of the antibacterial properties was performed in accordance with ISO 22196:2011 (measurement of antibacterial activity on plastics and other non-porous surfaces). The analysis was carried out in five experiments for each sample. Mueller Hinton Agar (MHA) was sterilized by autoclaving at (121 ± 2) °C, pressure (10^3^ ± 5) kPa for 30 min. A total of 20 mL of warm medium was poured into Petri dishes (Nuns, Denmark) and allowed to cool at ambient temperature. Then, *Staphylococcus aureus* (*S. aureus*, ATCC 25,923 strain) was cultured in beef extract-peptone (BEP) at 37 °C for 24 h and adjusted to a concentration of 10^7^ and 10^6^ CFU/mL. Next, 0.4 mL of *S. aureus* suspension was inoculated on the samples and covered with a piece of parafilm (4 cm × 4 cm). The Petri dishes containing inoculated samples (three samples of each type) were incubated at 37 °C for 24 h. After that, the parafilm pieces were transferred from the test samples to sterile tubes containing 9 mL of 0.9% NaCl for 1 h. After exposure, 1 mL of saline was transferred to the nutrient medium and incubated at 37 °C for 24 h. Finally, the number of CFU was counted.

## 3. Results

### 3.1. Search for Optimal Ag ALD Conditions

The most important task in the preparation of coatings by ALD is the selection of optimal conditions, which ensure that a sufficient amount of reagents is achieved for reactor saturation and self-limiting reactions. The required amount of reagent was achieved by changing the duration of the reagents’ inlet (pulse time) and the temperature of the reagent evaporator. The higher the evaporator temperature, the higher the reagent vapor pressure and the more reagents arrive at the substrate surface. In this regard, at the first stage of the search for optimal ALD conditions, the effect of the Ag(fod)(PEt_3_) evaporator temperature was studied (see Appendix A). According to ellipsometry and XRR data, no coating growth was observed at temperatures of 115,125, and 130 °C. At the evaporator temperature of 140 °C and 700 ALD cycles, the film thickness achieved at least 2 nm. A further increase in temperature led to a decrease in the growth rate. This is caused by the thermolysis of the precursor. Kariniemi et al. showed that Ag(fod)(PEt_3_) becomes unstable under high temperatures [16]. For the optimal ALD growth of Ag using Ag(fod)(PEt_3_), the temperature should be in the range of 120–160 °C [17]. Therefore, the reagent must not be heated above 150 °C.

The dependence of the coating growth rate on the reactor temperature was further investigated (Figure 1a). The optimal reactor temperature range is 155–165 °C, where the growth rate is maximum and relatively constant. The obtained ALD window for our set-up coincides with the upper threshold of the ALD window for similar processes described in the literature [17]. Our results indicated that at temperatures above 165 °C, the silver nanoparticles stop growing. At the same time, lowering the reactor temperature requires lowering the evaporator temperature, which leads to a decrease in the vapor pressure of the Ag(fod)(PEt_3_) precursor. Thus, the optimal conditions for the synthesis of silver NPs by ALD for our equipment are the evaporator temperature of 140–150 °C and the reactor temperature of 165 °C.

In view of the above, it was necessary to find another way to increase the amount of Ag(fod)(PEt_3_) entering the reactor to achieve optimal conditions for silver synthesis. The literature analysis showed [24] that it is useful to conduct several successive pulses of Ag(fod)(PEt_3_) in one ALD cycle. In our study, we tested regimes with 1, 3, 5, and 7 pulses of 4 s each (Figure 1b, normal pulses). In addition, regimes with 5, 7, and 11 pulses of Ag(fod)(PEt_3_) were used in the mode of the additional pressure increase in the Ag(fod)(PEt_3_) evaporator (Figure 1b, boost pulses). The purge time between Ag(fod)(PEt_3_) pulses was 2 s. The purging time after the last pulse in the ALD cycle was 10 s. It followed from the obtained results that saturation was achieved at 5 pulses. Furthermore, in the boost mode, the smallest thickness gradient was obtained. Using the boost mode, in turn, slightly increases the growth rate. Thus, it was found that to obtain the desired silver coatings, the synthesis should be performed in the boost mode using at least 5 Ag(fod)(PEt_3_) pulses of 4 s duration in one ALD cycle and reactor and evaporator temperatures of 165 and 150 °C, respectively. For further work, these synthesis conditions were used.

### 3.2. Study of Composition, Morphology and Structure of ALD Ag Nanoparticles

The morphology of Ag nanoparticles deposited on silicon and titanium substrates using 2300 ALD cycles was studied by SEM (Figure 2). The sample surface was characterized by the presence of a layer of evenly distributed individual nanoparticles. The Ag particle density on the titanium surface was noticeably higher than that on silicon (insets in Figure 2). Moreover, silver particles deposited on silicon have a narrower size distribution. The vast majority of the particles deposited on silicon were 20–28 nm in size, while the diameter of the particles deposited on the titanium surface varied in the range of 16–30 nm.

According to grazing incidence X-ray diffraction (GIXRD) data (Figure 3), the Ag ALD nanoparticles deposited on silicon are polycrystalline and characterized by (111), (200), (220), and (311) structure reflections of Ag (JSPDS 04-0783). Unfortunately, owing to the non-ideal surface plane, bulge and roughness of Ti support and Ti reflections that overlap the diffraction from silver, no Ag reflections were detected for the Ag NPs deposited there.

The results of the XPS analysis of the surface and bulk part of nanoparticles after the sputtering of the surface layer by argon ions showed that the deposited nanoparticles contain silver and impurities of carbon, oxygen, phosphorus and fluorine (Table 1). These data are consistent with the literature data. In most studies on the ALD of silver using Ag(fod)(PEt_3_), the presence of such impurity elements in a small amount is explained by the residues of incompletely reacted precursor ligands [25]. For the samples deposited on the titanium support, a noticeable amount of titanium was also found, which indicates that the layer of silver nanoparticles is non-conformal.

A more detailed study of the high-resolution XPS spectra showed that C1s spectra of surface (Figure 4a) and bulk (Figure 4d) measured after 30 s sputtering consist of 3 clear components. The most intensive peak at BE 285 eV corresponds to the aliphatic hydrocarbons (C–C, C–H), and the less intensive peak at BE 286.6 eV corresponds to the hydroxyl and C–O groups. The third low-intensity peak corresponds to the carboxyl and aldehyde groups. Most of the carbon originated from surface contamination during the storage of the samples in the air. However, the change in the carbon content during etching (Figure 4a,d) allows us to assume that a significant part of the carbon is the residues of the Ag(fod)(PEt_3_).

The O1s spectra of samples without sputtering (for surface) (Figure 4b) and after 30 s sputtering (for bulk) (Figure 4e) consist of 3 components, but their intensities differ significantly. The surface has low-intensity TiO_2_ and C=O/O–C=O components and high-intensity C–O/C–OH/Ti–OH components. After sputtering, the peak of TiO_2_ increased significantly.

The deconvolution of the Ag3d peaks (Figure 4c,f) indicated that the surface and bulk of Ag NPs consist of metallic silver. However, there is also a small amount of the oxide phase and Ag-F, Ag-P, Ag-C species.

### 3.3. Study of Composition, Morphology and Structure of Samples Deposited for Antibacterial and In Vitro Study

For the study of antibacterial properties and in vitro cell response of osteoblast-like osteosarcoma cells and fetal mesenchymal stem cells, the following four types of samples were prepared:(1)Polished titanium (Ti);(2)Titanium with ALD silver NPs (Ti-Ag);(3)Titanium with ALD titanium oxide nanocoatings (Ti-TiO_2_);(4)Titanium with ALD titanium oxide nanocoatings and silver NPs (Ti-TiO_2_-Ag).

The deposition of titanium oxide was carried out by the ALD according to a well-known technique [26], which also was described earlier in this paper [27]. Titanium chloride (IV) and water were used as precursors at 200 °C and 400 ALD cycles. The coatings obtained under these conditions were polycrystalline with an anatase structure (see Appendix A).

The morphology of the samples for in vitro experiments was studied by SEM. The initial surface of polished titanium is smooth and without any particles or defects (Figure 5a). For the Ti-Ag samples obtained after 2000 ALD cycles, silver NPs with a mean diameter of about 16–22 nm are observed on the surface (Figure 5b). The surface of the Ti-TiO_2_ samples (Figure 5c) is characterized by a continuous layer of titanium oxide with large (more than 50 nm) and small (20–30 nm) particles. Both large titanium oxide particles and small silver nanoparticles are observed on the surface of the Ti-TiO_2_-Ag samples (Figure 5d). The composition of the coatings was confirmed by SEM-EDS elemental analysis (Appendix A).

To assess the hydrophilicity and surface free energy (SFE) for the samples studied, the contact angles were measured using water (Figure 6a) and diiodomethane (Figure 6b). Based on the obtained data, calculations of the total surface free energy and its polar and dispersive components were carried out (Figure 6c). The surface of polished titanium is weakly hydrophilic (the water contact angle is about 80°) and has a small value of the polar component of the SFE. The atomic layer deposition of titanium oxide on titanium led to a decrease in the value of surface energy due to a decrease in the polar component and, hence, to the hydrophobicity increase (Θ = 91°), while the ALD of the silver resulted in a slight increase in the SFE and in the surface hydrophilicity (Θ = 69°). It is noteworthy that autoclaving, used to disinfect the material prior to in vitro study, led to the surface hydrophilicity increase for the titanium and titanium with an ALD TiO_2_ nanolayer, due to a significant increase in the polar component of SFE (Figure 6c). At the same time, for the samples containing ALD silver, the changes observed after autoclaving the samples were minimal. For all the samples studied, the CA values for weakly polar diiodomethane were close and practically did not change after autoclaving (Figure 6b).

The influence of autoclaving on the chemical composition of the surface was studied using XPS (Figure 7). An analysis of the high-resolution spectra of C1s (Figure 7a,d) and O1s (Figure 7b,e) showed that autoclaving leads to a significant increase in the number of polar hydroxyl groups, which is observed from a sharp increase in peaks near 286.6 eV at the C1s and 531.8 eV at the O1s spectra. Moreover, autoclaving reduced the amount of non-polar C–C and C–H groups of aliphatic hydrocarbons and, thus, increased surface free energy and hydrophilicity. It should also be noted that, according to the Ti2p spectra (Figure 7c,f), titanium was predominantly in the 4^+^ (TiO_2_) oxidation state. In addition, based on the absence of characteristic metallic Ti peaks in the region of 453–454 eV, one can conclude that the resulting TiO_2_ films are conformal.

### 3.4. Study of Antibacterial and Cytological Response

An analysis of the bactericidal properties of the samples was carried out using a strain of *Staphylococcus aureus*. All the samples, including the polished titanium without ALD structures, revealed antibacterial activity (Table 2). Significant bactericidal ability was found for the samples with a TiO_2_ layer, which reduced the initial number of CFU by 4–5 orders of magnitude. The Ti-Ag samples with silver NPs showed higher antibacterial activity. At the same time, the samples of the Ti-TiO_2_-Ag series demonstrated the best antibacterial ability.

Cell morphology after 24 h of cultivation was studied using scanning electron microscopy (Figure 8). The spreading of MG-63 cells occurred for all the samples (Figure 8a–d). On the surface of the polished titanium and titanium with an ALD TiO_2_ layer, the number of cells was the most significant. The number of cells on titanium with silver (Ti-Ag) was much lower. The cells on the Ti-TiO_2_-Ag sample surface (titanium oxide with silver NPs) were characterized by the formation of a large number of appendages filopodia. The cells are “connected” to each other by these filopodia (inset in Figure 8d). They are also visible on the Ti-TiO_2_ sample surface, but their number is smaller.

FetMSCs on the surface of all the samples are elongated, bipolar, spindle-shaped, and interconnected by filopodia (Figure 8e–h). This morphology of the cells is typical for the cells of this line [28,29]. The greatest number of cells was on the surface of the polished titanium. Round particles with a diameter of about 10 μm are visible on the surface of the Ti-TiO_2_ sample. They are evenly distributed over the entire surface of the sample (Appendix A). SEM-EDS showed that these particles consist of phosphorus, oxygen and calcium. We assume that they were formed as a result of crystallization from the phosphate buffer, which was used to stabilize the cells before studying the morphology.

After co-incubation of the MG-63 and FetMSCs, the cytotoxicity was tested using an MTT assay. All samples did not cause toxic activity (Figure 9). No significant difference in the viability for all the samples, including the control, was observed.

The differentiation of cells in the osteogenic direction was studied using early differentiation (alkaline phosphatase, ALP) and late differentiation (osteopontin, OPN) markers [23]. A medium that induces cell differentiation in the osteogenic direction was used. A similar trend of change in ALP activity was observed for MG-63 cells for all the samples with (red and orange bars) and without (grey and cyan bars) this medium (Figure 10a). The ALP activity for 1-week samples was higher than 2-week samples, indicating completion of early differentiation. However, a different trend was observed for FetMSCs (Figure 10b). In this case, the ALP activity increased after 2 weeks of cell cultivation (Figure 10a).

The induction medium more than doubles the activity of ALP for FetMSCs, but only slightly affects that for osteosarcoma MG-63 cells. At the same time, the effect of the medium for MG-63 cells increases for the samples with ALD structures (Ti-Ag, Ti-TiO_2_, Ti-TiO_2_-Ag) in comparison with the control sample and the sample of polished titanium (Ti). In general, the absolute values of the ALP activity of the samples are noticeably higher than those of the control sample; however, the difference between polished titanium and ALD samples is insignificant. Only the Ti-TiO_2_-Ag sample can be noted, whose activity was most affected by the differentiation-induced medium.

When cultivating MG-63 and FetMSCs on all the samples and controls with and without the induction medium, a gradual and significant increase in the OPN expression was observed throughout the entire period of cell cultivation (Figure 11). The OPN expression after the 3rd week compared with the 1st week using the induction medium increased by 6–18 times, depending on the sample and cell types. A multiple increase in osteopontin expression was also observed without the use of the medium. The greatest increase in OPN expression with the use of a differentiation medium was observed for the Ti-TiO_2_-Ag sample for both MG-63 and FetMSCs. In addition, a significant positive effect of the medium (more than 100%) was observed for the Ti-TiO_2_ samples during the cultivation of MG-63. For FetMSCs, a noticeable positive effect occurred after 3 weeks of culture for all the ALD samples, and also after the 2nd week for the Ti-TiO_2_-Ag samples.

In general, the OPN expression is higher for the ALD samples compared to the control (cultivation on plastic surface), which indicates a positive effect of ALD coatings on cell differentiation in the osteogenic direction.

## 4. Discussion

Conformal titanium oxide nanocoatings, silver NPs, and structures combining coatings and NPs have been successfully prepared by ALD. According to a number of in vitro and in vivo studies, it is known that ALD titanium oxide films have a positive effect on the proliferation, differentiation and osseointegration of titanium implants [19,30,31]. According to our results, as-deposited ALD TiO_2_ coatings have a less hydrophilic surface than polished titanium. This fact should have a negative effect on the osseointegration of the material, since many important biomolecules are weakly adsorbed on a hydrophobic surface [4], while a hydrophilic surface, on the contrary, enhances cell adhesion, proliferation and differentiation [32,33]. However, our results show that the standard pretreatment material for in vitro study (autoclaving) significantly increases surface wettability by increasing the concentration of hydroxyl groups on the surface. Thus, the hydrophobicity of the as-deposited TiO_2_ coatings should not adversely affect the rate of osseointegration. In this regard, it is noteworthy that the deposition of silver slightly increased the surface hydrophilicity, while autoclaving did not affect surface free energy and water wetting. Probably, the reason for this stability is the hydrogen plasma treatment during the Ag nanoparticle synthesis.

The results on a detailed study of the composition of the ALD Ag NPs by XPS showed that they contain a small amount of oxygen, carbon, fluorine, and phosphorus in the form of precursor residues. Similar results for ALD coatings were obtained earlier by other researchers [16,24,25]. However, the presence of such impurities does not noticeably affect the antibacterial properties of silver [34]. Indeed, ALD Ag NPs have shown high antibacterial activity against *Staphylococcus aureus*, which is one of the main causes of rejection of orthopedic implants [35]. However, the bactericidal activity of ALD Ag NPs was not significantly higher than that of the ALD titanium oxide coatings. The combination of titanium oxide coatings and Ag NPs showed the highest antibacterial activity, but in this case, it was most likely a cumulative effect rather than a synergistic one. However, the material with the combination of TiO_2_ coatings and Ag NPs has great prospects to prevent bacterial adhesion and biofilm formation, which can cause orthopedic implant-related infection after surgery. Once the biofilm has formed on the surface, it is extremely difficult to treat the infection even with a large dose of antibiotics.

Despite the excellent antibacterial properties, there are serious problems and a number of controversial issues regarding the use of silver for medical implants. On the one hand, the Ag-related production of ROS may have a negative effect on the viability, proliferation, and differentiation of preosteoblast cells. On the other hand, the rate of osseointegration of the implant significantly increases in the presence of an antibacterial effect simultaneously with the absence of preosteoblasts’ cytotoxicity, due to the “race for surface” effect [36]. Namely, preosteoblasts win the competition with bacteria to attach to the implant surface and proliferate. Despite the inconsistency of the results, it can be concluded that the cytological response is highly dependent on the amount of silver on the implant surface and the rate of its dissolution and leaching.

The results of the study of cell morphology at an early stage of cultivation (24 h) suggested that there is a negative effect of silver NPs on the early-stage adhesion of MG-63 and FetMSCs. However, the MTT test did not show any cytotoxicity for all the samples using both types of cells, and the Ti-TiO_2_-Ag sample showed a high level of filopodia formation, which is very important for cells’ later adhesion and migration [37]. Moreover, results of the in vitro study using both early and late differentiation markers showed the positive effect of Ag nanoparticles on osteogenic differentiation. It is especially important that differentiation in the osteogenic direction occurs even without the use of a differentiation-stimulating medium. At the same time, the rate of increase in ALP and OPN for both cell lines in the absence of an inducing medium was significantly higher for the ALD samples than in the control. Therefore, we believe that osteogenic differentiation is caused not only by the influence of cultivation conditions and/or medium but also by the influence of ALD titania and silver coatings.

A difference was found in the early differentiation of MG-63 and FetMSCs. The results showed that early differentiation is completed within the first two weeks of culture for osteoblast-like MG-63 cells. For mesenchymal stromal cells, an increase in ALP activity is observed at 2 weeks of cultivation, which indicates an incomplete stage of early differentiation. Despite the intrinsic osteogenic induction, osteosarcoma cells require much less time to complete early differentiation than mesenchymal stem cells.

It is worth noting the samples containing a combination of a nanolayer of titanium oxide and silver nanoparticles. These samples showed the highest number of filopodia and the most active differentiation both with and without induction medium, as well as the best antibacterial ability.

To date, there are many contradictory studies regarding the influence of silver nanoparticles on osteoblast-like and MSCs differentiation. Sengstock et al. have found that Ag NPs mitigate the osteogenic differentiation of human MSCs even at nontoxic concentrations, without affecting chondrogenesis [38]. The incorporation of Ag in TiO_2_ coatings exhibited negative effects on early MC3T3-E1 preosteoblast adhesion (within 2 h), mainly due to the toxicity of Ag ions [39]. However, cell differentiation according to ALP activity was relatively high. Samberg et al. have reported no influence of Ag NPs on human adipose-derived stem cell differentiation [40]. Rajendran et al. [41] have demonstrated the biocompatibility of Ag/TiO_2_ anatase structures using MG-63 and MSCs and bactericidal activity against *S. aureus*. At the same time, a stimulating effect of silver on the osteogenic differentiation of MSCs was found. Such a dramatic difference in the results can be explained by the significant influence of the silver concentration on the cytological response [42]. Recently, it has been established that the adherence and viability of the MG-63 and MC3T3-E1 cell differentiation are very sensitive to changes in Ag ion concentration [39].

Nowadays, various nanostructures based on titanium oxide and silver NPs have been studied. In some cases, ALD was used to obtain such structures, but only for the production of one of the materials. In most cases, titanium oxide [30,31,43] was deposited, but there are also studies on silver NPs deposited on nanotubes of various sizes [34]. In this respect, the approach proposed in our work is unique, because it uses only ALD to produce complex multicomponent material. In the future, we plan to test the possibility of controlling the characteristics of NPs and TiO_2_ layers over a wide range on complex-shaped substrates. In addition, the mechanical characteristics of coatings, such as adhesion, resistance to delamination and tribocorrosion, will be studied. ALD coatings are bonded to the substrate by strong chemical bonds, have low internal stresses and are usually very resistant to mechanical stress. However, their mechanical performance is highly dependent on thickness, composition and crystallinity [44,45,46], which requires further study for our coatings.

We believe that the prospects for the commercial application of our approach are quite high. Despite the low cost-effectiveness in terms of the use of precursors, ALD technology has found industrial application, due to its ability to deposit uniform coatings on the surface of a large number of supports simultaneously. For ALD, the complexity and shape of the surface do not play an important role. Therefore, ALD has been actively used in microelectronics and electronics for many years. There are a number of successful cases of the use of ALD in the industrial production of coatings for dental implants [47,48].

## 5. Conclusions

Ag nanoparticles, conformal TiO_2_ nanocoatings, and their combination have been successfully prepared by atomic layer deposition. Ag NPs contain small amounts of oxygen, carbon, fluorine, and phosphorus as Ag(fod)(PEt_3_) residues and increase the wettability of titanium surface. The ALD of TiO_2_ using TiCl_4_ and H_2_O leads to an increase in the surface hydrophobicity, while autoclaving the sample makes the surface more hydrophilic, due to the formation of surface hydroxyl groups.

Antibacterial properties against *S. aureus* were found for silver NPs and titanium oxide nanocoatings. However, the best ability was demonstrated for the samples with a combination of TiO_2_ nanocoatings and Ag NPs. These samples also showed the highest number of intercellular contacts of osteoblast-like MG-63 cells. All the obtained samples are non-cytotoxic for MG-63 and FetMScs, and the differentiation of cells proceeds in the osteogenic direction, both in the presence and absence of a differentiation-inducing medium. ALD biofunctionalization with TiO_2_ nanocoatings and Ag NPs is suggested to be a promising strategy to prevent infections and accelerate osseointegration.

## Figures and Tables

**Figure 1 jfb-13-00062-f001:**
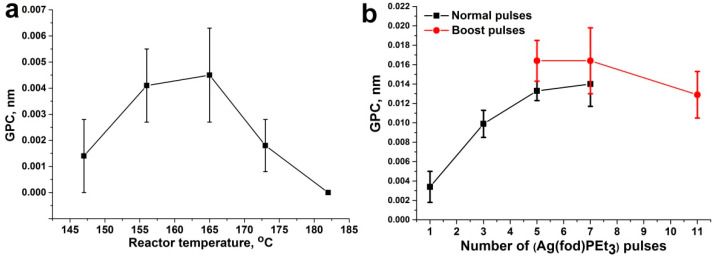
Growth per cycle (GPC) as a function of the reactor temperature (**a**) and number of Ag(fod)(PEt_3_) pulses in one ALD cycle (**b**). Error bars indicate the thickness gradient.

**Figure 2 jfb-13-00062-f002:**
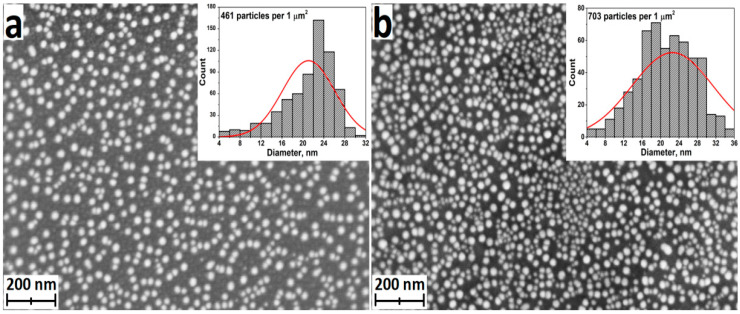
SEM images of Ag nanoparticles deposited on the silicon surface (**a**) and titanium disks (**b**). Insets demonstrate the diameter distribution histograms of the Ag nanoparticles.

**Figure 3 jfb-13-00062-f003:**
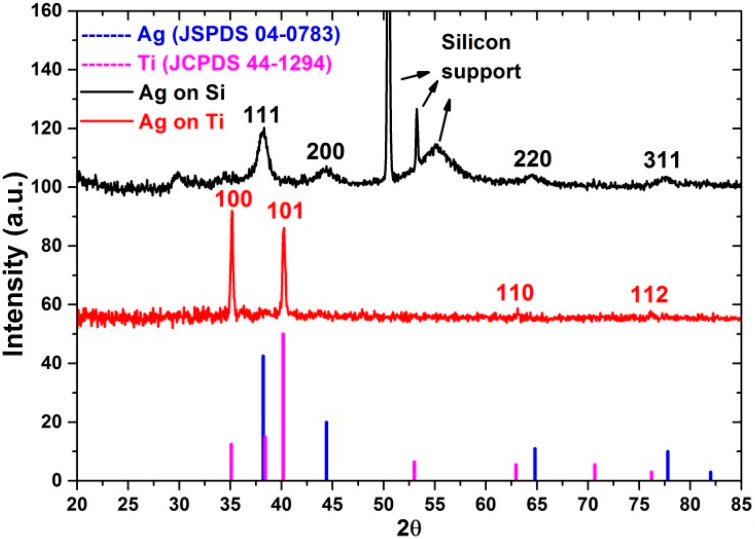
GIXRD patterns of the ALD Ag nanoparticles deposited on the silicon and titanium surfaces.

**Figure 4 jfb-13-00062-f004:**
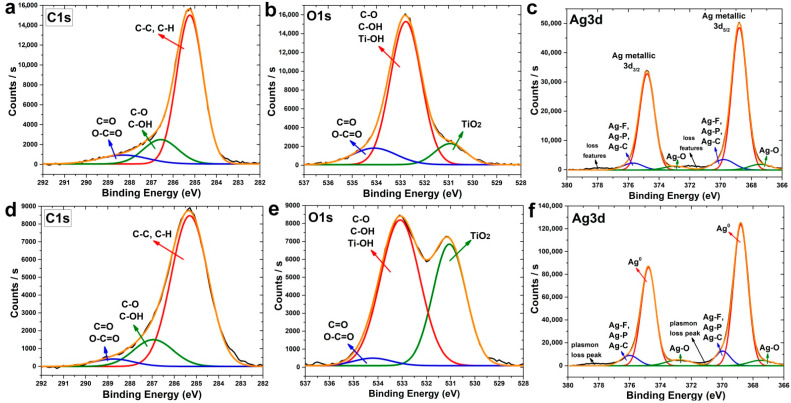
XPS spectra of C1s (**a**,**d**), O1s (**b**,**e**) and Ag3d (**c**,**f**) for ALD Ag nanoparticles before (**a**–**c**) and after 30 s Ag ion sputtering (**d**–**f**).

**Figure 5 jfb-13-00062-f005:**
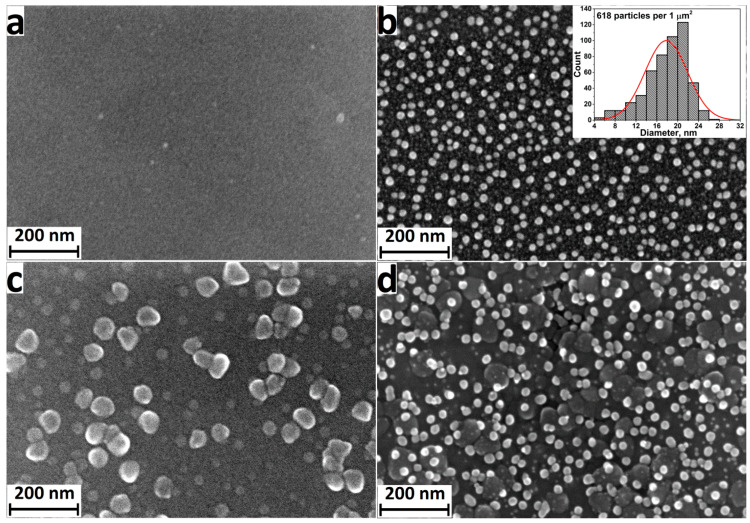
SEM images of surface for Ti support (**a**), Ti-Ag sample (Ag nanoparticles deposited using 2000 ALD cycles) (**b**), Ti-TiO_2_ sample (TiO_2_ coatings deposited using 400 ALD cycles) (**c**), and Ti-TiO_2_-Ag sample (**d**). Inset shows the diameter distribution histograms of the Ag nanoparticles.

**Figure 6 jfb-13-00062-f006:**
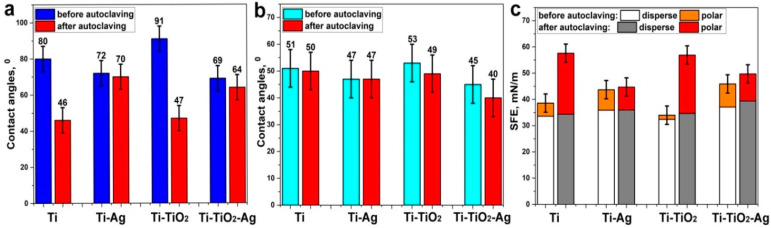
Results of water (**a**), diiodomethane contact angle measurements (**b**) and surface free energy calculation (**c**).

**Figure 7 jfb-13-00062-f007:**
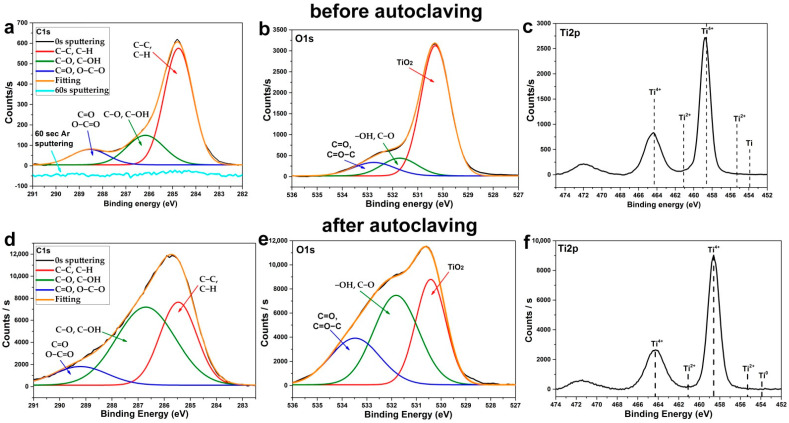
XPS spectra of C1s (**a**,**d**), O1s (**b**,**e**) and Ti2p (**c**,**f**) for ALD TiO_2_ coatings before (**a**–**c**) and after autoclaving (**d**–**f**) the Ti-TiO_2_ sample.

**Figure 8 jfb-13-00062-f008:**
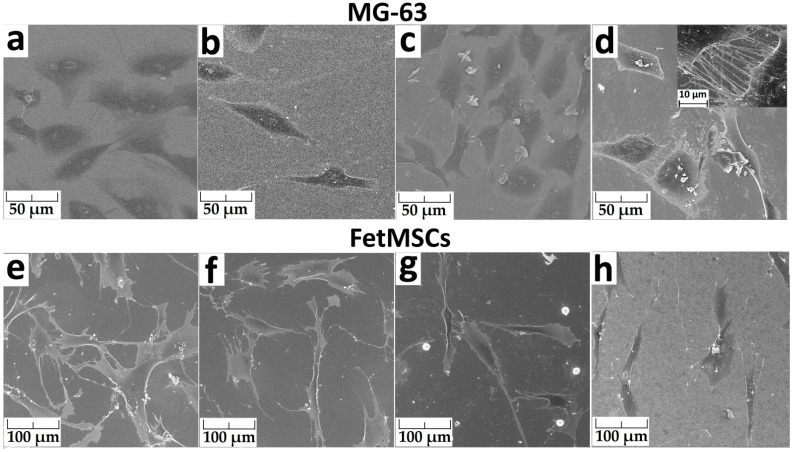
SEM images of MG-63 (**a**–**d**) and FetMSCs (**e**–**h**) co-incubated after 24 h of cultivation on Ti (**a**,**e**), Ti-Ag (**b**,**f**), Ti-TiO_2_ (**c**,**g**) and Ti-TiO_2_-Ag (**d**,**h**) samples.

**Figure 9 jfb-13-00062-f009:**
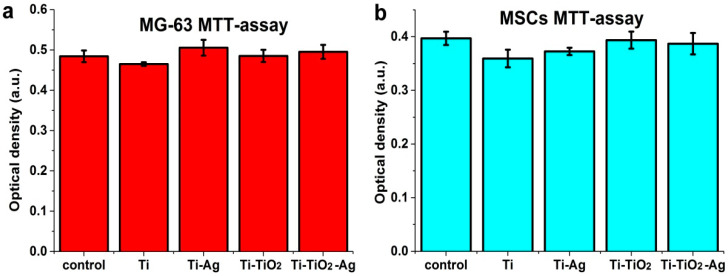
MG-63 (**a**) and FetMSCs (**b**) viability after co-incubation during 24 h. Data are presented as mean ± C.I. from five independent series of experiments (*p* < 0.05).

**Figure 10 jfb-13-00062-f010:**
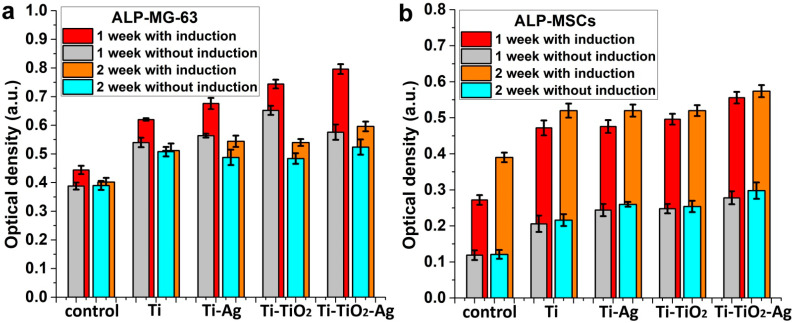
Alkaline phosphatase production by MG-63 (**a**) and FetMSCs (**b**) on the samples with and without a medium inducing cell differentiation in the osteogenic direction. Red and orange marks the values obtained with the induction medium. Each value represents mean ± C.I. from five independent experiments (*p* < 0.05).

**Figure 11 jfb-13-00062-f011:**
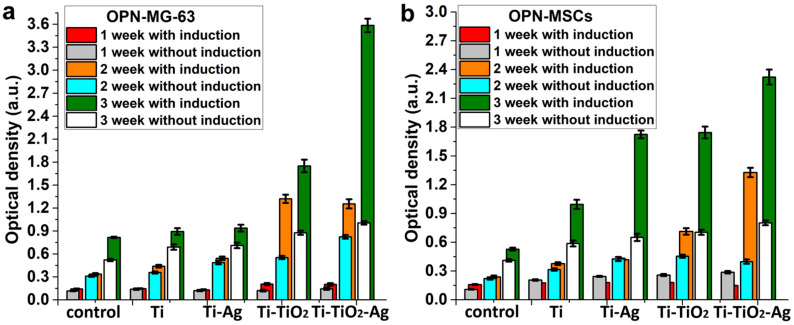
Osteopontin production by MG-63 (**a**) and FetMSCs (**b**) on the samples with and without a medium inducing cell differentiation in the osteogenic direction. Red, orange and green marks the values obtained with the induction medium. Each value represents mean ± C.I. from five independent experiments (*p* < 0.05).

**Table 1 jfb-13-00062-t001:** Chemical composition of ALD Ag nanoparticles determined by XPS.

Time of Sputtering, s	C, %	O, %	Ag, %	Ti, %	F, %	P, %
0	59.46	24.76	10.45	2.74	1.89	0.70
30	42.92	21.21	25.34	7.29	2.04	1.20
60	2.73	25.56	3.43	68.28	0	0

**Table 2 jfb-13-00062-t002:** Results of antibacterial study of samples using *Staphylococcus aureus*.

Sample	107 CFU/mL	106 CFU/mL
Ti	~10^5^	~10^4^
Ti-TiO_2_	~280	~150
Ti-Ag	~140	~80
Ti-TiO_2_-Ag	~85	42

## Data Availability

The main data had been provided in the article and Appendix A. Any other raw/processed data required to reproduce the findings of this study are available from the corresponding author upon request.

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
