# Peer review of "Antibacterial and Osteogenic Properties of Ag Nanoparticles and Ag/TiO_2_ Nanostructures Prepared by Atomic Layer Deposition"

_jfb, 2022, doi:10.3390/jfb13020062_

Round 1
Reviewer 1 Report
- Line number 66, page 2, the authors used the word ‘bacterial inflammation’ as a reason for most of implant failures. It should be either bacterial infection or the inflammation caused by bacterial infection.
- Did the authors study the scratch resistance property of these ALD Ti-TiO2-Ag disks? Since ALD is a coating technique, delamination of the deposited layers under load bearing applications is a common issue. Therefore, it is ideal to see the adhesion strength of this coating to the base metal before claiming it as an implant material.
- Page #.4; line 81: ‘line cells’ should be rectified.
- Page #. 4; line 152: Should be imaged, not imagined.
- Page # 4; line 184: Harvested is not the correct term here. Cells must have been maintained in culture, not harvested.
- Section 2.3.2: The statement concerning, “samples were autoclaved in water media” needs clarification.
- It was not clear, whether the authors have carried out the release study for the silver nanoparticles deposited on Ti surface? Considering that Ag would show concentration dependent toxicity and unless Ag is leaching out from the sample surface, there is no point in using the conditioned media for MTT assay. Direct seeding of cells on the material surface and then checking the cell viability may as well give the information pertaining to biocompatibility.
- Page #.4; line 188/189: Sentence rephrasing is required.
- Section 2.3.2: What was the dimension of dish used? It should be specified, since the seeding density mentioned was 5x106 cells/ml and medium volume was 300µl/well. Moreover, the reason for fixing the cells for 3 days for SEM studies was not clear. The authors should explain this.
- 1: To avoid confusion, there should have been consistency in describing evaporator temperature, deposition temperature and reactor temperature.
- The thickness of Ag coating mentioned in sections 2.2 and 3.2 are different.
- The Ag3d spectrum in Fig. 4c and f are missing.
- Section 3.2: As per the statement, “No Ag reflection -----non-ideal surface plane and roughness of Ti support”. In case no Ag nano-particles deposition was detected, how was the variation in response noted between Ti and Ti-Ag samples shown in Fig. 6 and Fig. 7-11?
- Table 2: Has the expt. been carried out only once? Besides, the usage of overlaid parafilm for CFU count may be erroneous. Considering the antibacterial activity of Ag coating is manifested by contact or upon its release, the authors could rather have tried to measure either the zone of inhibition by placing the Ti- samples after spreading the bacterial culture on the solidified growth medium followed by incubation, or grow the bacteria in presence of Ti- samples and subsequently take the inoculums to spread on the solidified medium and calculate the number of colonies following incubation.
- The % of Ag used could not be discerned. Since Ag will impart cytotoxic effect based on the concentration used, the same could have been mentioned clearly.
- Figure 8: The number of cells adhered on the Ag containing samples seem to be less compared to that seen in case of respective controls. It would have been ideal to keep the magnification/scale bars uniform in all the samples for proper comparison. Wherever necessary, the magnified view could have been shown as inset (as shown in Fig. 8d).
- What was the doubling time of FetMSCs used in the experiments, and based on which the authors presumed that the round particles seen in Fig. 8g are dividing cells? Moreover, it was also not apparent whether or not they have seen comparatively better cell growth on Ti-TiO2 surface to infer this. Else, they could have stained with any nuclear stain (ex: DAPI) to assess the cellular identity of the round structure.
- What was the rationale regarding choosing 24h incubation period for all the in vitro experiments with cells and why not longer?
- Details on statistical significance should have been given in the figures showing the quantitative data (for example: Fig. 9, 10).
- Fig. 11a: OPN seems to be higher at 3wk time point without osteoinduction (green bar). In case the labeling for white and green bar has been reversed, the authors should rectify the error. They need to include the statistical significance too in the figure.
- Pg. 13; Line 457: Should have been Ag-deposited instead of as-deposited.
- The authors may provide rationale / explanation backing their statement on autoclaving increasing the hydrophilicity property.
- Reference: Ref. # 41 and 42 are the same.
Author Response
Thank you very much for your careful review, questions, and comments. They really helped improve the manuscript. Unfortunately, when submitting the manuscript, we were in a hurry and it contained many errors and typos. Now we have carefully reviewed it and finalized it in accordance with the advice of the reviewers.
All of your questions are answered point by point. Together with the answers, we provide quotations from the text of the manuscript with corrections made. They are marked in italics.
- Line number 66, page 2, the authors used the word ‘bacterial inflammation’ as a reason for most of implant failures. It should be either bacterial infection or the inflammation caused by bacterial infection.
Corrected. Thank you.
- Did the authors study the scratch resistance property of these ALD Ti-TiO2-Ag disks? Since ALD is a coating technique, delamination of the deposited layers under load bearing applications is a common issue. Therefore, it is ideal to see the adhesion strength of this coating to the base metal before claiming it as an implant material.
High adhesion and resistance to delamination are very important for implant coatings. However, studies of these characteristics were not included in the objectives of our study at this stage. We have focused on аntibacterial and osteogenic properties and the effect of surface morphology and composition on them.
However, we can confidently assume that our coatings have very good mechanical characteristics.
First, ALD provides excellent adhesion of coatings because the growth of coatings is due to surface chemical reactions between the substrate and ALD precursors. Therefore, the coating and the substrate are connected by very strong chemical bonds. Good adhesion and delamination resistance are confirmed by many studies [1-3]. However, these characteristics are very dependent on the composition and crystal structure of the coating.
The good interface of our coatings can be judged from high-resolution TEM data, which can be downloaded from the link: https://disk.yandex.ru/i/K16n2RIsb5ZMUQ.
Secondly, high delamination and cracking are usually characteristic of rather thick films of micron thickness. Our films are ultra-thin - about 20-30 nm. However, further research is needed to confirm this.
In order to reflect the issue of mechanical properties in the text of the manuscript, we have added the following phrases to the penultimate paragraph of the discussion section:
In addition, it is necessary to study the mechanical characteristics of coatings such as adhesion, resistance to delamination and tribocorrosion. ALD coatings are bonded to the substrate by strong chemical bonds, have low internal stresses, and are usually very resistant to mechanical stress. However, their mechanical performance is highly dependent on thickness, composition and crystallinity [44-46], which requires further study for our coatings.
1. Kilpi, L.; Ylivaara, O.M.E.; Vaajoki, A.; Malm, J.; Sintonen, S.; Tuominen, M.; Puurunen, R.L.; Ronkainen, H. Microscratch testing method for systematic evaluation of the adhesion of atomic layer deposited thin films on silicon. Journal of Vacuum Science & Technology A: Vacuum, Surfaces, and Films 2016, 34, doi:10.1116/1.4935959.
2. Ylivaara, O.M.E.; Langner, A.; Liu, X.; Schneider, D.; Julin, J.; Arstila, K.; Sintonen, S.; Ali, S.; Lipsanen, H.; Sajavaara, T.; et al. Mechanical and optical properties of as-grown and thermally annealed titanium dioxide from titanium tetrachloride and water by atomic layer deposition. Thin Solid Films 2021, 732, doi:10.1016/j.tsf.2021.138758.
3. Radi, P.A.; Testoni, G.E.; Pessoa, R.S.; Maciel, H.S.; Rocha, L.A.; Vieira, L. Tribocorrosion behavior of TiO2/Al2O3 nanolaminate, Al2O3, and TiO2 thin films produced by atomic layer deposition. Surface and Coatings Technology 2018, 349, 1077-1082, doi:10.1016/j.surfcoat.2018.06.036.
3. Page #.4; line 81: ‘line cells’ should be rectified.
Corrected. Thank you.
4. Page #. 4; line 152: Should be imaged, not imagined.
Corrected. Thank you.
5. Page # 4; line 184: Harvested is not the correct term here. Cells must have been maintained in culture, not harvested.
Corrected. Thank you.
6. Section 2.3.2: The statement concerning, “samples were autoclaved in water media” needs clarification.
Thank you very much for this clarifying question. We were inaccurate with the wording and translation into English. We used water fluid steam sterilization at 121 C and a pressure of 1.5 atmospheres
We have added this information to section 2.4.2.
7. It was not clear, whether the authors have carried out the release study for the silver nanoparticles deposited on Ti surface? Considering that Ag would show concentration dependent toxicity and unless Ag is leaching out from the sample surface, there is no point in using the conditioned media for MTT assay. Direct seeding of cells on the material surface and then checking the cell viability may as well give the information pertaining to biocompatibility.
Thank you for this important question. Ag release was not studied by us. However, similar ALD silver coatings on nanotubes have been studied by our colleagues earlier [4]. A slow Ag release was observed. In our case amount of Ag release will be not enough for reliable measurements, because the surface of our samples is flat, and they had nanotubes with a very high specific surface area. To confirm the release, we would have to make special samples with a large surface. We considered it inappropriate.
You are right. It is very likely that the lack of cytotoxic effect in our MTT assay is due to the very low silver release. But it is still an important result. Roughly speaking, we have shown that our coating will not have a cytotoxic effect on surrounding cells and tissues. But at the same time, silver shows a “contact” antibacterial effect. That is, the formation of bacterial biofilms on the surface of the implant is unlikely, as well as the ingress of bacteria living on the surface of the implant into the body.
Unfortunately, the other MTT test methods are hardly possible for our samples. The MTT-test involves measuring the optical density of the solution, but our titanium supports are non-transparent, and it is impossible to measure the optical density directly. The other method which uses incubation with subsequent washing of cells gives a very high error since not all cells are washed off due to high adhesion and not all survive these procedures.
4. Radtke, A.; Jedrzejewski, T.; Kozak, W.; Sadowska, B.; Wieckowska-Szakiel, M.; Talik, E.; Makela, M.; Leskela, M.; Piszczek, P. Optimization of the Silver Nanoparticles PEALD Process on the Surface of 1-D Titania Coatings. Nanomaterials (Basel) 2017, 7, doi:10.3390/nano7070193.
8. Page #.4; line 188/189: Sentence rephrasing is required.
Corrected. Now this sentence looks like this:
Then the samples were placed into the wells of 4-wells plates (Nunc, США). MG-63 and FetMSCs cells were seeded (1*105 in 20 µl of DMEM/F12 nutrient medium) on the surface of the samples and maintained for 24 hours in CO2.
9. Section 2.3.2: What was the dimension of dish used? It should be specified, since the seeding density mentioned was 5x106 cells/ml and medium volume was 300µl/well. Moreover, the reason for fixing the cells for 3 days for SEM studies was not clear. The authors should explain this.
Thanks a lot. We mixed up the methods for studying morphology and differentiation. For morphology, the seeding of 5x106 cells in 20µl medium was used. For the differentiation study also 300µl/well medium was added to fully cover the samples. We have rewritten the sections 2.4.2. Cell morphology and 2.4.4. Cells osteogenic differentiation analysis
3 days of fixation: Indeed, there was no need to fix samples for 3 days. This is not part of our methodology. However, cultivation and sample preparation for SEM was carried out at different institutes and time was needed for transportation. It is likely that long exposure to a phosphate buffer and glutaraldehyde led to the formation of round particles and flakes, which are clearly visible in Figure 8 c and g. SEM-EDS showed that these particles and flakes are consisted of P, O, and Ca.
We discuss this in more detail when answering your question 15
10. 1: To avoid confusion, there should have been consistency in describing evaporator temperature, deposition temperature and reactor temperature.
Thank you. Indeed, you can get confused if you are not an expert in ALD. Deposition temperature and reactor temperature are synonyms. Now we have chosen to use only the terms – “reactor temperature” and “evaporator temperature”. These are two different things. The text was corrected.
11. The thickness of Ag coating mentioned in sections 2.2 and 3.2 are different.
There is no mistake here. I believe that you meant not thickness but the number of ALD cycles. In 2.2 was indicated data for TiO2 (not Ag) and in 3.2 for Ag. For the deposition of titanium oxide, 400 cycles were used, and for the deposition of silver, from 350 to 2300. The studies of the effect of temperature, the pulse times the reagents on growth rate were used 350, 400, 700 cycles (see table S1).
To study the morphology and size of Ag nanoparticles we have to use more ALD cycles (2000 and 2300).
2000 cycles were used for in vitro study.
In order to avoid confusion for the reader, we made clarifications in section 2.2 - The total number of TiO2 ALD cycles was 400. The total number of Ag ALD cycles varied from 350 to 2300.
Also in section 3.2 (first paragraph): The morphology of Ag nanoparticles deposited on silicon and titanium using 2300 ALD cycles was studied by SEM.
We hope that now you and any other readers will not have any confusion.
12. The Ag3d spectrum in Fig. 4c and f are missing.
Thanks a lot. Yes, indeed, I mixed up the Ti2p and Ag3d spectra. Now the figure is correct.
13. Section 3.2: As per the statement, “No Ag reflection -----non-ideal surface plane and roughness of Ti support”. In case no Ag nano-particles deposition was detected, how was the variation in response noted between Ti and Ti-Ag samples shown in Fig. 6 and Fig. 7-11?
The deposition of silver on Ti supports has indeed taken place. There can be no doubt. We can see silver peaks on the XPS spectra (fig 4c,f), and on the EDS spectra locally in different parts of the surface (fig S2,3). We finally see the particles in the SEM images (fig 2b, 5b,d)
The reason why these particles are visible on silicon XRD and not visible on titanium XRD patterns is quite obvious. In order to make XRD of such thin layers, grazing angle XRD (GIXRD) is used. This is the only possible way to measure such small objects. In this case, the x-ray rays are directed along the surface plane of samples at a very small angle. As a result, if the substrate is uneven, or has a bulge, the signal from the substrate overlaps the signal from the coating. Even an unevenness of a few micrometers will completely block the signal from particles of a couple of tens of nanometers. Silicon wafers are almost perfectly even and still we see very intense peaks from silicon, but polishing titanium evenly is really a very difficult task.
To make it more clear to the reader, we have edited the text:
“No Ag reflection was detected for Ag NPs deposited on titanium due to non-ideal surface plane, bulge and roughness of Ti support which overlaps the diffraction from silver.”
- Table 2: Has the expt. been carried out only once? Besides, the usage of overlaid parafilm for CFU count may be erroneous. Considering the antibacterial activity of Ag coating is manifested by contact or upon its release, the authors could rather have tried to measure either the zone of inhibition by placing the Ti- samples after spreading the bacterial culture on the solidified growth medium followed by incubation, or grow the bacteria in presence of Ti- samples and subsequently take the inoculums to spread on the solidified medium and calculate the number of colonies following incubation.
Indeed, the Kirby-Bayer method (Disk diffusion test) may seem very effective. We started our research with this method. But, alas, the data on the disk-diffusion method were not reproducible and zones of inhibition were very small. We also tested the technique of placing samples in a liquid inoculum with different concentrations of bacteria: from 10*8 cfu/ml to 10*5 cfu/ml. We observed growth suppression by 30-40% due to the high concentration of microbial suspension, but this, in our opinion, is not enough.
The method with parafilm showed the best results. We conducted a series of studies with coatings of various types, which were reproducible in triplicate. The data about a number of experiments is presented in section “2.5. Antibacterial activity”
- The % of Ag used could not be discerned. Since Ag will impart cytotoxic effect based on the concentration used, the same could have been mentioned clearly.
Yes, you are right, this is a very important factor. In order to emphasize this, we have added the following explanation to the discussion (Discussion section, 3rd paragraph):
Despite the inconsistency of the results, it can be concluded that the cytological response is highly dependent on the amount of silver on the surface of the implant and the rate of its dissolution and leaching.
- Figure 8: The number of cells adhered on the Ag containing samples seem to be less compared to that seen in case of respective controls. It would have been ideal to keep the magnification/scale bars uniform in all the samples for proper comparison. Wherever necessary, the magnified view could have been shown as inset (as shown in Fig. 8d).
Now all images have the same magnification.
From the presented figures, one can indeed conclude that there are fewer cells on the Ti-Ag surface. However, for cell morphology there is not much difference. Based on these data, no unambiguous conclusions about Ag effect can be drawn, because only one sample was used for SEM study. For reliable conclusions, we need to make several samples and accumulate statistics.
- What was the doubling time of FetMSCs used in the experiments, and based on which the authors presumed that the round particles seen in Fig. 8g are dividing cells? Moreover, it was also not apparent whether or not they have seen comparatively better cell growth on Ti-TiO2 surface to infer this. Else, they could have stained with any nuclear stain (ex: DAPI) to assess the cellular identity of the round structure.
This was a mistake. Indeed you are right. These are not dividing cells. After the submission of the manuscript, we had the opportunity to make an elemental microanalysis of these particles. They mainly contain phosphorus, oxygen, and calcium. As we wrote in the answer to question 9, the samples were kept for quite a long time (3 days) in phosphate buffer with glutaraldehyde. As a result, these particles were formed. Also, the flakes with similar compositions were observed on the surface of the same sample (Ti-TiO2) with MG-63 cells (Figure 8c). Probably the surface of the Ti-TiO2 sample promotes the mineralization of calcium-phosphate which is very important for bone mineralization. However, additional studies are needed to confirm this, so we will not discuss this issue in the revised version of the manuscript.
SEM-EDS showed that these particles consist of phosphorus, oxygen and calcium. We assume that they were formed as a result of crystallization from the phosphate buffer, which was used to stabilize the cells before morphology studying.
- What was the rationale regarding choosing 24h incubation period for all the in vitro experiments with cells and why not longer?
24h incubation periods were used only for viability (MTT-test) and cells morphology study by SEM. For differentiation assays 7, 14, and 21 days of cultivation were used. Moreover, for the morphology study, we used also 2h cultivation, but this time was not enough for good adhesion. In our experience, if you use a time of more than 24 hours, then the cells will grow and merge into a monolayer and all samples will look the same. If we want to catch the differences, 24 hours is enough.
For the MTT test, a time in the range of 1-3 days is usually used, because it is enough to assess the cytotoxicity of cells. We have chosen 24 hours as it was expected that already during this period silver cytotoxicity could manifest itself and it was possible to associate the obtained data with cell morphology.
- Details on statistical significance should have been given in the figures showing the quantitative data (for example: Fig. 9, 10).
Thank you. Description of the statistical analysis was added to the experimental part:
2.4.5. Statistical Analysis.
Three samples of each type were used for MTT-test and five samples for differentiation analysis. The error bars in the figures represent the confidence interval (CI). Student’s t tests were used to evaluate the differences between the experimental and control groups. Credible interval P <0.05 was considered statistically significant for all tests.
The figure captions contain data about confidence interval (CI) and credible interval - P
- Fig 11a: OPN seems to be higher at 3wk time point without osteoinduction (green bar). In case the labeling for white and green bar has been reversed, the authors should rectify the error. They need to include the statistical significance too in the figure.
Yes, it's a typo. We have corrected the figure. Thanks a lot!
- page 13; Line 457: Should have been Ag-deposited instead of as-deposited.
Corrected. Thank you
- The authors may provide rationale / explanation backing their statement on autoclaving increasing the hydrophilicity property.
It’s a very good question. It is known that the hydrophilicity of TiO2 is very unstable. The adsorbed molecules and surface chemical groups (OH, O, CH3, etc.) influence greatly wettability and surface energy. There are a lot of articles that show the effect UV treatment, aging, temperature, and humidity on TiO2 wettability.
In our case, water vapor at high pressure and temperature was used. As a result, the amount of hydrophilic hydroxyl (OH) groups increase but hydrophobic C-C, C-H decrease. This is confirmed by XPS and described in the manuscript between fig 6 and fig 7.
It is most likely that the aliphatic hydrocarbons adsorbed during storage in air, and evaporated at autoclaving. At the same time, the water vapor hydroxylated the surface. But this is just a guess. For confirmation, a very complex study is necessary. In our work, it was important to show that the hydrophobicity of the titanium oxide surface should not adversely affect practical applications such as coatings for implants. Since autoclaving is necessary before implantation, the surface will become hydrophilic in any case. We discuss this in detail at the beginning of the discussion section.
In order to make the reasons for the change in hydrophilicity more understandable for the reader, we have added a clarifying sentence in the paragraph between fig 6 and fig 7:
Thus, autoclaving increases surface energy and hydrophilicity by reducing the number of hydrophobic C-C and C-H groups and increasing the number of hydrophilic –OH
- Reference: Ref. # 41 and 42 are the same.
Corrected. Thank you!!!

Reviewer 2 Report
The manuscript reported Ag/TiO2 nanostructured film prepared by atomic layer deposition and its antibacterial and osteogenic properties. The ALD deposition of pure silver nanoparticles is novel, at least for me. The manuscript was elle written, except some language or format problems.
‘Then the samples were placed in MG-63 and FetMSCs line cells were inoculated (5*106/ml) on the surface of the samples for 24 hours in CO2.’
‘This is caused by the thermolysis of the precursor which according to the literature Ag(fod)(PEt3) becomes unstable under these temperatures [16].’
‘The data obtained indicate that we cannot raise the reactor temperature above 165 °C, because silver stops growing, but on the other hand, we cannot reduce it, since it is necessary to reduce the source temperature, and there will not be sufficient vapor pressure of Ag(fod)(PEt3).’(better passive, too long).
‘2.3.2. Cell morphology. Titanium samples were autoclaved in water media prior to in vitro studies.’ I wonder if the contents of F and P were changed (elemental ratios in xps) by the autoclaving in water, for the removal of precursor residual and the relaesed silver ions are vital for cell behaviors.
Table 2, ‘10*7 10*6 KOE/MA’, please explain.
Author Response
Thank you very much for your review, questions and comments. They really helped improve the manuscript. All of your questions are answered point by point. Together with the answers, we provide quotations from the text of the manuscript with corrections made. They are marked in italics.
The manuscript reported Ag/TiO2 nanostructured film prepared by atomic layer deposition and its antibacterial and osteogenic properties. The ALD deposition of pure silver nanoparticles is novel, at least for me. The manuscript was elle written, except some language or format problems.
‘Then the samples were placed in MG-63 and FetMSCs line cells were inoculated (5*106/ml) on the surface of the samples for 24 hours in CO2.’
Thank you, we have corrected the text:
Then the samples were placed into the wells of 4-wells plates (Nunc, США). MG-63 and FetMSCs cells were seeded (1*105 in 20 µl of DMEM/F12 nutrient medium) on the surface of the samples and maintained for 24 hours in CO2.
‘This is caused by the thermolysis of the precursor which according to the literature Ag(fod)(PEt3) becomes unstable under these temperatures [16].’
Thank you, we have corrected the text:
This is caused by the thermolysis of the precursor. Kariniemi et al. showed that Ag(fod)(PEt3) becomes unstable under these temperatures [16].
‘The data obtained indicate that we cannot raise the reactor temperature above 165 °C, because silver stops growing, but on the other hand, we cannot reduce it, since it is necessary to reduce the source temperature, and there will not be sufficient vapor pressure of Ag(fod)(PEt3).’(better passive, too long).
Thank you, we have corrected the text:
The results indicate that we cannot raise the reactor temperature above 165 °C, since the silver stops growing. But we can't lower it either, because for this we have to lower the evaporator temperature and decrease Ag(fod)(PEt3) vapor pressure.
‘2.3.2. Cell morphology. Titanium samples were autoclaved in water media prior to in vitro studies.’ I wonder if the contents of F and P were changed (elemental ratios in xps) by the autoclaving in water, for the removal of precursor residual and the relaesed silver ions are vital for cell behaviors.
Thank you very much for this clarifying question. We were inaccurate with the wording and translation into English. It was not “autoclaving in water media”. We used water fluid steam sterilization at 121 C and a pressure of 1.5 atmospheres. We have added this information to the section 2.4.2.
We did not study the change in the concentration of phosphorus and fluorine after autoclaving, but it would hardly have changed under steam treatment conditions. Although even in this case, their effect would be negligible, since their initial concentration in very small (20 nm) silver particles does not exceed 1-2%. The cytological effect of silver should be much greater.
Table 2, ‘10*7 10*6 KOE/MA’, please explain.
This is a typo. Cyrillic characters were used. KOE/ml is correct. Thank you!

Reviewer 3 Report
The manuscript introduced a type of TiO2/Ag nanostructures fabricated via ALD technique, and systematic investigations on the antibacterial properties of the resulting samples. The results are presented with a reasonable analysis and enough data. Thus, the referee considers that the submitted results will give readers a valuable aid point and further opportunity for the investigations in this field. Therefore, the reviewer considers the manuscript is of fine quality & scope for the publication in Journal of Functional Biomaterials after a minor revision.
Comment 1. Author tried temperatures of the reagent bottle at 115,125,130 °C, and then suddenly jumped to 150 °C. How about 130 - 150 °C? As mentioned by author, the optimized temperature can be appeared within this range.
Comment 2. The size distribution and density histograms of Ag nanoparticles should be added in the revised version (i.e. Nanoscale, 2018, 10, 22737, Nano-Micro Lett. (2020) 12:114).
Comment 3. EDS maps are required to verify element distribution for the Ti-TiO2-Ag sample (i.e. Small 2019, 1901606).
Comment 4. Why did author decide use ALD technique for the fabrication of Ag nanoparticles? Additional discussions should be added in the introduction section.
Author Response
Thank you very much for your review, questions and comments. They really helped improve the manuscript. All of your questions are answered point by point. Together with the answers, we provide quotations from the text of the manuscript with corrections made. They are marked in italics.
The manuscript introduced a type of TiO2/Ag nanostructures fabricated via ALD technique, and systematic investigations on the antibacterial properties of the resulting samples. The results are presented with a reasonable analysis and enough data. Thus, the referee considers that the submitted results will give readers a valuable aid point and further opportunity for the investigations in this field. Therefore, the reviewer considers the manuscript is of fine quality & scope for the publication in Journal of Functional Biomaterials after a minor revision.
Comment 1. Author tried temperatures of the reagent bottle at 115,125,130 °C, and then suddenly jumped to 150 °C. How about 130 - 150 °C? As mentioned by author, the optimized temperature can be appeared within this range.
Thank you for this question. In fact, we used a step of no more than 10 degrees. You helped us find the typo. In the second row of Table S1, the evaporator temperature was indicated as 150C, but in fact it was 140. This mistake led to a mistake in the text. Indeed, the optimal temperature is not 150, but about140C.
It is also worth noting that a more accurate determination of the optimum temperature is not very important. When changing the temperature of the reagent bottle, it is important to reach a temperature sufficient for the good evaporation of the reagent. But the amount of the reagent entering the reactor and the saturation conditions are achieved by varying the time of the reagent pulse.
Comment 2. The size distribution and density histograms of Ag nanoparticles should be added in the revised version (i.e. Nanoscale, 2018, 10, 22737, Nano-Micro Lett. (2020) 12:114).
Thanks for your advice. We added the diameter histograms of Ag nanoparticles and density calculation data in fig 2 and 5. Unfortunately, we failed to obtain a reliable histogram for the Ti-TiO2-Ag sample, because the surface of the sample is uneven and, therefore, the contrast of the particles is inhomogeneous. In addition, the surface, contains titanium oxide particles, which affects the result.
Also we added accompanying text:
Section 3.2. 1st paragraph: “The Ag particle density on the titanium surface is noticeably higher than on silicon (Figure 2 insets). Moreover, silver particles deposited on silicon have a narrower size distribution. The vast majority of the particles deposited on silicon have a size of 20-28 nm, and the diameter of the particles deposited on the surface of titanium varies in the range of 16-30 nm”
Section 3.2. 2nd paragraph: On the surface of the Ti-Ag sample (Figure 5b), obtained after 2000 ALD cycles, silver NPs with a mean diameter of about 16-22 nm are visible.
Comment 3. EDS maps are required to verify element distribution for the Ti-TiO2-Ag sample (i.e. Small 2019, 1901606).
Thanks for your advice. We tried to make a EDS map of the elements distribution. However, this was a meaningless, because the maps show only noise. The resolution in SEM-EDS rarely reaches 100 nm, but the size of our silver particles is about 20 nm. Titanium oxide particles are slightly larger but do not exceed 60 nm. In the article that you recommended, the size of the structures is much larger than 100 nm, so it makes sense to make a map.
Instead of a distribution map, we made SEM-EDS measurements in several random areas of the surface and confirmed the uniform distribution of silver, oxygen and titanium on the surface of the Ti-TiO2-Ag sample. These data are presented in Figures S3 and Table S2 in the supplementary file.
We hope that this data can be an equivalent replacement for the distribution map that shows an even distribution of elements.
Comment 4. Why did author decide use ALD technique for the fabrication of Ag nanoparticles? Additional discussions should be added in the introduction section.
There are several reasons – advantages of ALD:
1) High-quality coatings (purity, interface between the substrate)
2) The ability to easily adjust the thickness or size by changing the number of cycles
3) Ability to deposit uniform coatings on substrates of complex shape and a large number of substrates at the same time (this is very important for industrial production)
4) The ability to combine the deposition of silver and titanium oxide in one technological process, i.e. use one setup
We have rewritten the text in the introduction to clarify these advantages of ALD (page 2, the last paragraph):
The cyclical nature of the ALD processes provides precise thickness control down to the sub-nanometer. In addition, self-limiting surface reactions at the substrate-gas interface ensure that films grow in a layer-by-layer mechanism and allow for conformal deposition of thin films on complex three-dimensional and porous substrates. All these features and good scalability have led to an increase in industry interest for ALD and a dramatic increase in publications over the past ten years [15]. The possibility of using one technology for the synthesis of both continuous coatings of titanium oxide and silver NPs is also an important factor when choosing ALD as a method for deposition of biomedical coatings.

Round 2
Reviewer 1 Report
The authors have tried addressing most of the queries/concerns that were raised earlier. However, the followings may require further attention.
- Pg. 6; line 269-271: Providing the range for reactor temperature is fine. However, the sentence stating "the reagent cannot be heated above 110-150 °C" should be rephrased. Either they can simply mention the upper limit as 150 °C, and above this the heating should not be performed, or they should give the range (should not mention above, if giving range).
- Writing FetMSCs cells should be avoided, it should rather be mentioned as FetMSCs, since MSCs stands for mesenchymal stem cells.
- Pg. 16; line 471: What does it mean by number of OPN? Sentence rephrasing may be required.
- Despite poor/reduced cell adhesion in case of Ag containing ALD samples, the OPN seemed comparatively higher there. This would probably mean the influence of secretory factors underlying this (in here, Ag could be a plausible candidate), which would promote osteogenesis in cells growing in presence of the ALD samples. If this is to be believed, the concentration of Ag used and deposited on each, if not the release, needs to be specified in the methods and result sections. It may be better also to show cell adhesion on ALD samples after 48-96h of cell seeding (by seeding cells depending on the doubling time and possibly at a lower density).
- Pg.18; line 499: What is the meaning of as-deposited TiO2 coatings?
- The same reference has been mentioned twice as Ref # 41 and 42. It should be rectified.
- Overall, the authors may pay due attention to sentence framing. At some places incomplete sentences were also seen. Below are some of the examples on the same.
Incomplete sentence: (pg # 6, line # 263; Pg.18; line 547).
Sentence rephrasing may be required (Pg. 2; Line77; Pg. 10, line 352; Pg.16; line 460; Pg. 18; line 551).
Author Response
The authors have tried addressing most of the queries/concerns that were raised earlier. However, the followings may require further attention.
Thank you very much for your careful review, questions and comments.
All of your questions are answered point by point. Together with the answers, we provide quotations from the text of the manuscript with corrections made. They are marked in italics.
Also, English was extensively edited. Therefore, the .pdf file with the manuscript contains a corrected version in a clean form (corrected sentences are highlighted in yellow) and a version showing "all markup" made using the MS Word tracking function
- Pg. 6; line 269-271: Providing the range for reactor temperature is fine. However, the sentence stating "the reagent cannot be heated above 110-150 °C" should be rephrased. Either they can simply mention the upper limit as 150 °C, and above this the heating should not be performed, or they should give the range (should not mention above, if giving range).
Yes, thank you. We changed the sentence to "the reagent must not be heated above 150 °C"
- Writing FetMSCs cells should be avoided, it should rather be mentioned as FetMSCs, since MSCs stands for mesenchymal stem cells.
Yes, thank you. You are right. We have corrected the use of the term FetMSCs throughout the text.
- Pg. 16; line 471: What does it mean by number of OPN? Sentence rephrasing may be required.
Thank you, we meant the amount of OPN production (OPN expression). Now we have replaced it with «OPN expression»:
The OPN expression after the 3rd week compared with the 1st week using the induction medium increased by 6-18 times
- Despite poor/reduced cell adhesion in case of Ag containing ALD samples, the OPN seemed comparatively higher there. This would probably mean the influence of secretory factors underlying this (in here, Ag could be a plausible candidate), which would promote osteogenesis in cells growing in presence of the ALD samples. If this is to be believed, the concentration of Ag used and deposited on each, if not the release, needs to be specified in the methods and result sections. It may be better also to show cell adhesion on ALD samples after 48-96h of cell seeding (by seeding cells depending on the doubling time and possibly at a lower density).
Silver «secretory factors» could indeed influence differentiation. This is a reasonable explanation. However, ALD of crystalline titanium oxide also accelerated differentiation. It is possible that surface charge, wettability, or morphology were also determinants of differentiation (both silver and titanium oxide were present on the surface in the form of nanoparticles). Therefore, it is impossible to discuss this with certainty.
Regarding the concentration of silver, here we cannot write anything new. In the experimental part, we indicated the size of the substrates, and based on the SEM, we calculated the average particle size and density of their arrangement (this was advised to us by another reviewer – figures 2 and 5). We believe this is enough.
The «concentration of Ag used» cannot be determined because the precursor enters the reactor in the form of vapors and does not all precursor molecules react with the substrate.
The study of morphology after 48 and 96 hours of cultivation could indeed be helpful to discuss the results of the Ti-Ag sample. But for our studies, we used 2 and 24 hours, since there is still no noticeable proliferation and differentiation during these times. It was impossible to know in advance what results in we would get. Unfortunately, we cannot carry out these additional measurements in a short time, because to do this, we will have to prepare new substrates, deposit titanium oxide and silver on them, and only then carry out new in vitro tests.
- Pg.18; line 499: What is the meaning of as-deposited TiO2 coatings?
As-deposited means «freshly deposited» or is not subjected to additional processing (in our work, additional processing was autoclaving). This term is actively used for thin films and coatings. Since we are discussing wettability measurements in this part of the manuscript and autoclaving greatly affects wetting, we decided to use this term. So in our case, the term «as-deposited samples» was used as an antonym for samples after autoclaving
- The same reference has been mentioned twice as Ref # 41 and 42. It should be rectified.
Thank you! We corrected this error in the previous revision. But… Probably there was a bug of the Microsoft Word/EndNote interaction. Now we removed all EndNote references and made them by hand.
- Overall, the authors may pay due attention to sentence framing. At some places incomplete sentences were also seen. Below are some of the examples on the same.
Incomplete sentence: (pg # 6, line # 263; Pg.18; line 547).
We have edited the sentence framing:
pg # 6, line # 263. - In this regard, at the first stage of the search for optimal ALD conditions, the effect of the Ag(fod)(PEt3) evaporator temperature was studied
Pg.18; line 547. - It is worth noting the samples containing a combination of a nanolayer of titanium oxide and silver nanoparticles.
Sentence rephrasing may be required (Pg. 2; Line77; Pg. 10, line 352; Pg.16; line 460; Pg. 18; line 551).
We have rephrased the sentences:
Pg. 2; Line77 - Silver nanoparticles (NPs) have proved to be effective in medical implants because of their high surface area and antibacterial properties against gram-positive and gram-negative bacteria.
Pg. 10, line 352 - Deconvolution of Ag3d peaks (Figures 4c and 4f) indicated that the surface and bulk of Ag NPs consist of metallic silver.
Pg. 18; line 551 To date, there are many contradictory studies regarding the influence of silver nanoparticles on osteoblast-like and MSCs differentiation